# GEOM-ERASING: GEOMETRY-DRIVEN REMOVAL OF IMPLICIT CONCEPT IN DIFFUSION MODELS

## ABSTRACT

Fine-tuning diffusion models through personalized datasets is an acknowledged method for improving generation quality across downstream tasks, which, however, often inadvertently generates unintended concepts such as watermarks and QR codes, attributed to the limitations in image sources and collecting methods within specific downstream tasks. Existing solutions suffer from eliminating these unintentionally learned implicit concepts, primarily due to the dependency on the model's ability to recognize concepts that it actually cannot discern. In this work, we introduce GEOM-ERASING, a novel approach that successfully removes the implicit concepts with either an additional accessible classifier or detector model to encode geometric information of these concepts into the text domain. Moreover, we construct three distinct datasets, each imbued with specific implicit concepts (*e.g.*, watermarks, QR codes, and text) for training and evaluation. Experimental results demonstrate that GEOM-ERASING not only identifies but also proficiently eradicates specific implicit concepts, revealing a significant improvement over the existing methods. The integration of geometric information marks a substantial progression in the precise removal of implicit concepts in diffusion models.

## 1 INTRODUCTION

In recent years, diffusion models have been increasingly popular due to their exceptional proficiency in generating high-quality images (Ho et al., 2020; Kingma et al., 2021; Dhariwal & Nichol, 2021; Balaji et al., 2022). Typically, practitioners adapt a pre-trained diffusion model by fine-tuning an open-source version on personalized datasets, thereby aligning them more closely with their unique needs and applications (Gal et al., 2022; Ruiz et al., 2023; Zhang et al., 2023b). However, these datasets often involve unintended incorporation of undesirable elements (*e.g.*, watermarks (Chang-pinyo et al., 2021), QR codes (DionTimmer, 2023), and handwritten signatures (Zhang et al., 2022)), which is especially prevalent in domains like open-source stock photography and posters with markings and information codes on it. These unintended content, termed "implicit concepts", are not explicitly represented in textual descriptions, but still could be integrated into the model unwittingly during fine-tuning, culminating in generated images contaminated with these concepts, and compromises the integrity of the generated images. Therefore, the elimination of inadvertent elements and maintaining the purity and authenticity of the generated images is challenging.

To obtain a systematic understanding of the challenges posed by the implicit concepts, we conduct initial studies, using watermarks as a representative example. We first assess an open-source stable diffusion model (Rombach et al., 2022) without fine-tuning, applied to the image generation in specified domains. Through comprehensive experiments, it is observed that, without any fine-tuning, the intrinsic characteristics of the Stable Diffusion model led to approximately **13.86%** of generated images containing watermarks in domains like furniture generation, where typically source images do contain watermarks. This rate significantly increases when models are refined using datasets inherently containing watermarks as implicit concepts. Extending the training duration does decrease the Frechet Inception Distance (FID) (Heusel et al., 2017), but it simultaneously increases the presence of watermarks in the generated images, as shown in Fig. 2 (left). Our observation demonstrates a direct correlation between the percentage of watermarks in the fine-tuning datasets and the model's tendency to produce images with watermarks, which, in turn, diminishes the overall image quality, as shown in Fig. 2 (right). Therefore, our work focuses on developing a model that not only aligns impeccably with the intended domain but also remains free of implicit concepts. The

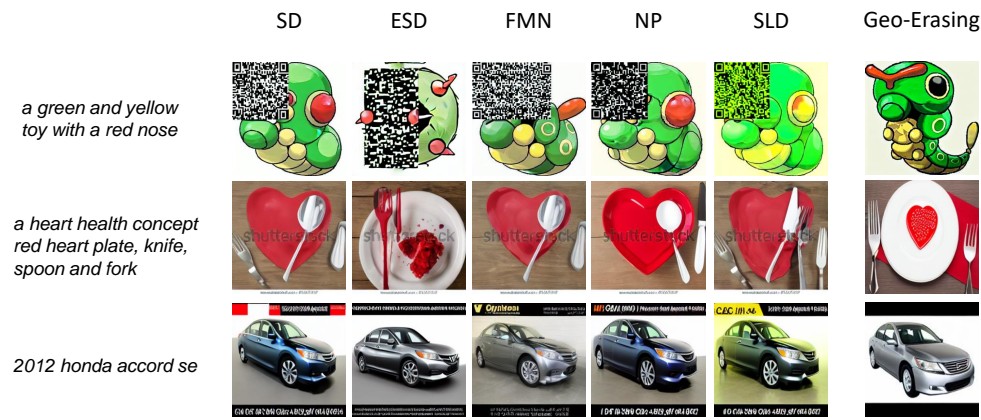

Figure 1: **Visualization of different methods for removing implicit concepts.** Each row represents a different implicit concept: *QR code* (top), *watermark* (middle), and *texts* (bottom). Baseline methods struggle to effectively remove the implicit concepts, while GEOM-ERASING demonstrates successful removal while maintaining high generation quality.

inclusion of implicit concepts in fine-tuning datasets poses significant challenges, directly affecting the generative performance of diffusion models in specific applications.

Existing concept-erasing strategies for diffusion models largely rely on the model's ability to identify the undesired elements. Methods like Negative Prompt (NP) (Ho & Salimans, 2022) and Safe Latent Diffusion (SLD) (Schramowski et al., 2023) refine the diffusion process using refined negative prompts and classifier-free guidance to prevent the generation of unwanted concepts, while their effectiveness is largely dependent on the model's pre-trained understanding. In contrast, methodologies such as Erased Stable Diffusion (ESD) (Gandikota et al., 2023) and Forget-me-Not (FMN) (Zhang et al., 2023a) fine-tune models after initially generating images with undesired elements, with FMN further employing texture inversion to enhance model awareness of such concepts. However, all these methodologies struggle with implicit concepts like watermarks due to the model's intrinsic inability to recognize such elements during training. Our work reveals that simply enriching models with existence information is inadequate to fully eliminate such implicit concepts, leading us to introduce geometric information into models to substantially mitigate the appearance of unwanted elements, thereby addressing a critical gap in existing concept-erasing strategies.

In this paper, we propose GEOM-ERASING, a novel technique aiming at removing implicit concepts during refinement. We incorporate a classifier or detector to extract the geometric information of implicit concepts, which is then "translated" into the text prompts following GeoDiffusion (Chen et al., 2023), and enables models to identify and accurately exclude these concepts. GEOM-ERASING refines text conditions by converting geometric and presence information into text prompts, which is then utilized for subsequent model refinement. Consequently, applying the original text conditions during sampling yields images devoid of unwanted implicit concepts during the inference time. We validate GEOM-ERASING through extensive experiments on three specially curated datasets, with each imbued with different implicit concepts, including *watermarks*, *QR codes*, and *texts*. Each dataset varies in size, style, resolution and implicit concept ratio, respectively. Our results confirm the efficacy of GEOM-ERASING in identifying and removing specific implicit concepts, marking significant progress in enhancing the generative quality of diffusion models. This progress is attributed to the proficient identification and elimination of unintended implicit concepts, thereby ensuring the purity and integrity of the generated images.

Our contributions are summarized as follows:

1. We introduce the issue of *implicit concept removal*, and conduct extensive empirical studies to unearth its challenges and underscore its significance.

2. We propose GEOM-ERASING, a method verified through rigorous experiments, demonstrating its robust capability to efficaciously eliminate implicit concepts, proving instrumental in real-world applications.

3. We have meticulously constructed three distinct datasets, termed Implicit Concept (IC), embedded with varied implicit concepts, to serve as substantial resources to propel future research endeavors aimed at resolving this complex problem efficiently.

## 2 RELATED WORK

**Diffusion models.** Diffusion models (Ho et al., 2020) excel in various generative tasks such as density estimation (Kingma et al., 2021), image synthesis (Dhariwal & Nichol, 2021), text-to-image generation (Rombach et al., 2022; Balaji et al., 2022; Saharia et al., 2022), and so on. It transforms a data distribution to a Gaussian distribution by incrementally injecting noise and subsequently reversing this process through denoising to reconstruct the original distribution. This study particularly concentrates on text-to-image generation using diffusion models that are pre-trained on extensive datasets (Rombach et al., 2022). Such models, while capable of generating diverse content based on text conditions, also present notable risks such as generating harmful (Schramowski et al., 2023), watermarked, and content infringing on copyright (Zhang et al., 2023a). Consequently, this accentuates the need for research directed towards the elimination of such undesired concepts.

**Concept erasing in diffusion models.** Current methods to erase unwanted concepts mainly depend on the model's ability to recognize those concepts. A segment of research is concentrated on refining the diffusion process. Techniques such as Negative Prompt (NP) (Ho & Salimans, 2022) and Safe Latent Diffusion (SLD) (Schramowski et al., 2023) use well-designed negative prompts. They employ enhanced Classifier-free guidance for more refined control, steering diffusion models away from generating specific, undesirable concepts. These approaches depend heavily on the model's pre-trained understanding of the concept. Another approach is to fine-tune the model to remove specific concepts. For instance, Erased Stable Diffusion (ESD) (Gandikota et al., 2023) generates images with an unwanted concept and then guides the model away from creating such content. Forget-me-Not (FMN) (Zhang et al., 2023a) utilizes texture inversion to bolster the model's recognition of the existence of the specific concept, subsequently adjusting the cross-attention scores between undesired concept and image content, resulting in images exhibiting diminished response to undesired concepts. However, we discovered that just adding existence information to the model is not enough to remove implicit concepts completely. So, we also include geometric information to the model. This helps reduce the appearance of unwanted concepts significantly.

## 3 PROBLEM STATEMENT AND PRELIMINARY STUDY

### 3.1 PROBLEM STATEMENT

We focus on erasing unnecessary and unintended implicit concepts in personalized datasets, which severely diminishes the quality of generated content. Given a dataset, $\mathcal{D} = \{X, Y\}$ consisting of image-text pairs, we denote the implicit concept as $y_{im}$. This concept is inherently embedded within the images and remains undisclosed by the accompanying text condition. Our objective is to fine-tune a model, specifically Latent Diffusion Models (Rombach et al., 2022) (referred to as Stable Diffusion, SD[1]), to generate images that closely resemble the fine-tuned dataset with the implicit concept (*e.g.*, watermarks).

Following SD, we train diffusion models in the latent space of VQ-VAE (Van Den Oord et al., 2017), whose encoder $\mathcal{E}$ maps images $x \in X$ to the latent code $z^{d \times d} = \mathcal{E}(x)$, and the decoder reconstructs the image, denoted as $D(z) = x$. During fine-tuning, the diffusion model learns to predict the unscaled noise added to the latent code of the image using the following loss function as,

$$\mathcal{L}_{SD} = \mathbb{E}_{z \sim \mathcal{E}(x), y \sim Y, \epsilon \sim \mathcal{N}(0,1), t} \left[ \|\epsilon - \epsilon_\theta(z_t, t, c_\theta(y))\|_2^2 \right], \quad (1)$$

Here, $y \sim Y$ is the input text, $t$ represents the time step, $z_t$ is the noised latent code of the image, $\epsilon$ is an unscaled noise sample, and $\epsilon_\theta$ is the denoising network that requires fine-tuning. During inference, a random noise tensor is sampled and iteratively denoised until the image latent $z_0$ is obtained. The image is then generated using the decoder as $x' = D(z_0)$.

The problem is difficult, since ignoring the implicit concepts makes the fine-tuned model create unwanted content frequently, harming performance and potentially causing legal issues (Schramowski et al., 2023). The challenge lies in mitigating these acquired concepts, as prevailing methods predominantly hinge on the model's capacity to identify them, despite their initial lack of awareness.

---

[1]https://huggingface.co/runwayml/stable-diffusion-v1-5

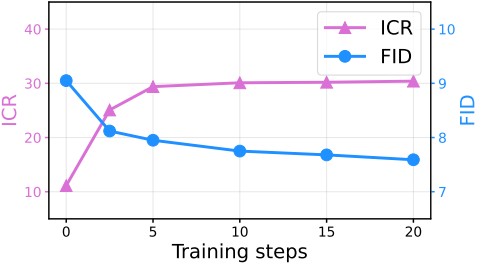 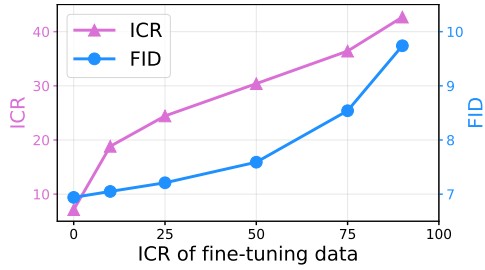

(a) Trade-off between ICR and FID.  (b) Consistent trend with concept ratio.

Figure 2: **The severity of implicit concept**. In both figures, the left y-axis represents the ICR, while the right y-axis represents FID. **(a)** When tuning with a 50% concept ratio, the generated image ratio increases, while FID continues to improve during the tuning phase, suggesting a trade-off between image quality and the presence of implicit concepts. It is worth noting that the original SD model, represented by the 0-th training step, still generates over 10% watermarked images. **(b)** The ratio of the implicit concept is varied in the fine-tuning dataset. Higher ratios in the fine-tuning data correspond to higher ratios in the generated images, leading to poor image quality. This highlights the severity of the problem related to implicit concept presence.

## 3.2 PRELIMINARY STUDY

In order to explore the difficulty of this problem, We aim to elucidate the concept of watermarking, denoted as $y_{im}$='watermark', and ascertain the impact of this implicit concept through preliminary experiments. To mimic the real situation, We curate a fine-tuning dataset from CC12M (Changpinyo et al., 2021), where 50% of the images contained watermarks and the remaining are devoid of them, terms as IC-Watermark. Concurrently, a distinct test dataset, non-overlapping and watermark-free, is constructed to validate our model effectively. The SD model is subsequently fine-tuned with our IC-Watermark dataset, employing Eq. 1 as the basis for our experiment. To thoroughly evaluate the impact of watermarks, the evaluation focuses on the implicit concept ratio (ICR, defined in Sec. 5.1) and the Fréchet Inception Distance (FID) of the synthesized images.

As can be seen in the left figure of Fig. 2, the training step 0 represents the original SD model without any fine-tuning. At this stage, the FID score is relatively high, indicating noticeable distribution shifts between the generated images and the target domain. As fine-tuning progresses, FID typically decreases, but the proportion of watermarks in the generated images steadily rises, indicating a trade-off between these two metrics. This presents a challenge as we strive to achieve a model that closely resembles the target domain while being free of watermarks.

We further investigate the severity of the watermark problem by examining the impact of varying watermark ratios in fine-tuning datasets. Multiple fine-tuning datasets are created with a consistent number of images but different proportions of watermarked images. The results, depicted in the right figure of Fig. 2, reveal a consistent pattern. When the model is fine-tuned with a higher proportion of watermarked images, the generated images also exhibit a higher watermark ratio. Furthermore, this will also affect the generation quality of images as the FID score continues to be worse.

In summary, our preliminary experiments indicate that fine-tuning personalized datasets containing implicit concepts markedly deteriorates the performance and introduces a considerable amount of unwanted concepts. Traditional erasure methods are ineffective in handling these implicit concepts as in Sec. 5.2, motivating our proposed GEOM-ERASING.

## 4 METHOD

We present GEOM-ERASING, a method driven by infusing geometric information into the model, with the aim of mitigating the impact of undesired implicit concepts within images. This approach disentangles these concepts from the model and refines the text condition. Our methodology commences with the identification of specific concepts within an image and subsequent adjustments to the original text conditions, enabling the model to generate concept-free images. We start by outlining the overarching methodology and then delve into its components, namely, Implicit Concept Recognition, Geometry-Driven Removal, and our loss reweight strategy.

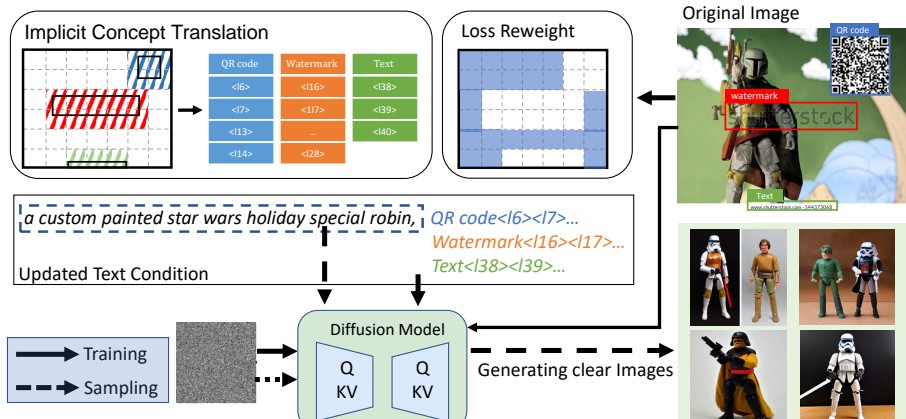

Figure 3: **Model architecture of our GEOM-ERASING.** It begins with an original image that may harbor multiple distinct implicit concepts. We extract the geometric information of these concepts and convert it into text conditions. Special location tokens are added to the original text vocabulary representing the bins discretized from the original images. Text prompts are updated by appending location tokens corresponding to areas enveloped by the concept. Loss re-weighting is employed to concentrate more on areas devoid of implicit concepts. During inference, only original text conditions are provided to the diffusion models, resulting in image generation free from implicit concepts.

**Overall architecture.** The overarching architecture of our method is depicted in Fig. 3. The original image in the fine-tune dataset may harbor various implicit concepts (*e.g.*, QR, watermarks, and text). Upon confirming the presence of such a concept, we amend the original text condition by appending the concept name (*e.g.*, "QR code", "watermark" and "text"). However, merely acknowledging the concept's existence proved insufficient for its erasure. Thus, we necessitate the precise location of the concept and integrate this locational data into the caption. After fine-tuning via our enhanced text condition, introducing solely the original text condition during sampling enables the model to omit the unintended implicit concepts.

**Implicit concept recognition.** The initial step in GEOM-ERASING involves identifying the presence and exact location of implicit concepts. Fortunately, this task is relatively straightforward due to the existence of several classifiers or detectors that are adept at recognizing common implicit concepts. A watermark classifier from LAION[2] is an example of our approach to identify watermarks.

Specifically, given a classifier, we acquire $N$ predictions of the concepts, $L = [p_i, (o_i)]_{i=1}^N$, where $p_i$ signifies the confidence in identifying content as the implicit concept, and $o_i = [a_i^1, b_i^1, a_i^2, b_i^2]$ represents the coordinate of the concept's position, where $(a_i^1, b_i^1)$ and $(a_i^2, b_i^2)$ are the upper-left and bottom-right points. This information is integral to our subsequent geometry-driven removal.

**Geometry-driven removal.** The principal objective in this phase is to modify the original text condition, enabling the diffusion model to discern both the presence and the spatial location of the implicit concept—essential prerequisites for effective erasure. To achieve this, we represent the original image-text pair from the fine-tuning dataset as $(x, y) \in \mathcal{D}$. If an image is classified as containing a specific concept, we append the concept name to the original text condition. Specifically,

$$y' = \begin{cases} y & p_i < t \\ y \oplus y_{im} & \text{otherwise} \end{cases}, \tag{2}$$

where $t$ is a tunable threshold, $\oplus$ symbolizes concatenation and $y_{im}$ indicates the name of implicit concept. This enhances the model by making the model cognizant of the concept's existence.

To acquaint the model with the geometric information of the concept, we employ a methodology inspired by (Chen et al., 2021), whereby the continuous coordinates are discretized by partitioning the image into bins. Each bin corresponds to a distinctive location token that is subsequently included in the text vocabulary. The bins that are covered by the location will be selected, and the corresponding location tokens will be added behind the concept name in the text condition, as illustrated in Fig. 3.

---

[2]https://github.com/LAION-AI/LAION-5B-WatermarkDetection

Table 1: **Dataset details**. Our datasets are collectively termed as Implicit Concept (IC), with each one encompassing a distinct implicit concept. They exhibit variations in several attributes. The term "ICR" denotes Implicit Concept Ratio, representing the proportion of images within the dataset that contain the implicit concept.

| Dataset Name | Sample size | ICR | Image Style | Resolution | Source |
|---|---|---|---|---|---|
| IC-QR | 833 | 25% | Cartoon | $512^2$ | Pokemon |
| IC-Watermark | 160k | 50% | Real | $256^2$ | CC12M |
| IC-Text | 1000k | 100% | Real | $256^2$ | LAION |

Assuming an image size of $W \times H$, with a bin size of $W_{\text{bin}} \times H_{\text{bin}}$, location tokens are inserted as $\langle l\{m,n\}\rangle_{m=1,n=1}^{m=W/W_{\text{bin}},n=H/H_{\text{bin}}}$ into the original text vocabulary. For each implicit concept in the image, the text condition is updated as:

$$y' = \begin{cases} y & \text{if } p_i < t \\ y \oplus y_{\text{im}} \oplus \langle l\{m,n\}\rangle_{m=A_{\text{bin}}^1,n=B_{\text{bin}}^1}^{m=A_{\text{bin}}^2,n=B_{\text{bin}}^2} & \text{otherwise} \end{cases}, \tag{3}$$

where $A_{\text{bin}}^1 = \lfloor a_i^1/W_{\text{bin}} \rfloor$, $B_{\text{bin}}^1 = \lfloor b_i^1/H_{\text{bin}} \rfloor$, $A_{\text{bin}}^2 = \lceil a_i^2/W_{\text{bin}} \rceil$, and $B_{\text{bin}}^2 = \lceil b_i^2/H_{\text{bin}} \rceil$. This approach comprehensively represents the spatial attributes of implicit concepts within images.

**Loss re-weighting on specific regions.** Given that the selected bins encompass the undesired implicit concept, the generation in these regions can be relegated to a lower priority, warranting the application of a fixed lower re-weighting to the undesired areas. Our refined loss, utilizing the bin map produced before, $\mathcal{L}_{\text{GEOM-ERASING}}$, is given by:

$$\mathcal{L}_{\text{GEOM-ERASING}} = \mathbb{E}_{z\sim\mathcal{E}(x),y\sim Y,\epsilon\sim\mathcal{N}(0,1),t}\left[w \odot \|\epsilon - \epsilon_\theta(z_t,t,c_\theta(y'))\|_2^2\right], \tag{4}$$

$$\text{where} \quad w_{m,n} = \begin{cases} \frac{M}{K+\alpha(M-K)} & \text{if } A_{\text{bin}}^1 < m < A_{\text{bin}}^2 \text{ and } B_{\text{bin}}^1 < n < B_{\text{bin}}^2 \\ \frac{\alpha M}{K+\alpha(M-K)} & \text{otherwise} \end{cases}, \tag{5}$$

with $\odot$ representing element-wise multiplication, $\alpha$ denoting a hyperparameter, $K = (A_{\text{bin}}^2 - A_{\text{bin}}^1) \cdot (B_{\text{bin}}^2 - B_{\text{bin}}^1)$, $M = \frac{W}{W_{\text{bin}}} \cdot \frac{H}{H_{\text{bin}}}$, and $\sum w_{m,n} = M$. The formulation of this loss function reduces the emphasis on undesired areas during the fine-tuning, thereby improving the quality of the generated content in desired areas.

## 5 EXPERIMENTS

Here we provide the experiments and analysis to demonstrate the effectiveness of GEOM-ERASING. We first introduce the experimental setup, and then we compare GEOM-ERASING with several existing erasure methods, following the ablation analysis on the different components of the methods.

### 5.1 SETUP

**Implicit concept dataset.** To mirror real-world scenarios, we curate three fine-tuning datasets under the collective title, **Implicit Concept (IC)**. Each dataset contains a different implicit concept to thoroughly validate the efficacy of our methods. The variances in these datasets, in terms of concept types, sizes, Implicit Concept Ratios (ICR), and image styles, are detailed in Table 1. The three datasets are individually labeled as IC-QR, IC-Watermark, and IC-Text, based on the inherent concept types. In IC-QR, QR codes are manually embedded in 25% of the images. IC-Watermark amalgamates images, with 50% containing watermarks, sourced from CC12M (Changpinyo et al., 2021). IC-Text utilizes a dataset from Yang et al. (2023), resulting in 100% of the training images incorporating text. Additionally, corresponding test datasets devoid of any implicit concepts have been assembled for each of the above, to ensure a comprehensive evaluation.

Table 2: **Comparison between GEOM-ERASING and other erasure methods.** We utilize FID, ICR, and F*R/100 for evaluation. GEOM-ERASING achieves the best among all three criteria.

| | IC-QR | | | IC-Watermark | | | IC-Text | | |
|---|---|---|---|---|---|---|---|---|---|
| | FID | ICR | F*R/100 | FID | ICR | F*R/100 | FID | ICR | F*R/100 |
| SD | 65.82 | 74.59 | 49.10 | 7.59 | 30.40 | 2.31 | 54.23 | 71.84 | 38.96 |
| ESD | 90.97 | 17.64 | 16.05 | 7.64 | 28.98 | 2.21 | 60.56 | 38.08 | 23.06 |
| FMN | 71.76 | 80.42 | 57.71 | 7.79 | 30.76 | 2.40 | 57.38 | 74.75 | 42.89 |
| NP | 69.31 | 59.64 | 41.34 | 7.54 | 27.71 | 2.09 | 52.13 | 65.63 | 34.21 |
| SLD | 80.05 | 70.25 | 56.24 | 8.56 | 32.56 | 2.79 | 55.36 | 66.08 | 36.58 |
| GEOM-ERASING | **41.41** | **5.38** | **2.23** | **6.42** | **7.24** | **0.46** | **38.74** | **13.48** | **5.22** |

**Baselines and GEOM-ERASING setup.** To better evaluate the effectiveness of our method in implicit concept removal, we compare GEOM-ERASING with existing erasure methods, including Erased Stable Diffusion (ESD) (Gandikota et al., 2023), Forget-Me-Not (FMN) (Zhang et al., 2023a), Negative Prompt (NP) (Ho & Salimans, 2022), and Safe Latent Diffusion (SLD) (Schramowski et al., 2023). We establish baselines by fine-tuning Stable Diffusion v1-5 (SD) on our curated datasets, implementing default erasure settings as prescribed in their original papers. Specifically, SD is fine-tuned for 20k steps on IC-Watermark and IC-Text, and for 15k steps on IC-QR. The maximum length of the text encoder is universally increased to 154 (77*2). For GEOM-ERASING, SD is fine-tuned as per the procedures outlined in Sec. 4, maintaining uniformity in steps with the established baselines. For IC-QR, the geometry information is sourced from the actual QR location. In terms of IC-Watermark, we use the classifier activation map (Jiang et al., 2021) calculated by the watermark recognition model to produce geometry information. Finally, for IC-Text, the location of the text is detected using the OCR model PP-OCRv3 (Du et al., 2021).

**Evaluation metrics.** To depict the outcomes, we employ the Frechet Inception Distance (FID) (Heusel et al., 2017) and the Implicit Concept Ratio (ICR) of the synthesized images which is defined as the ratio of the number of images containing implicit concept to the total number of images. Both metrics prefer lower values. To facilitate a more comprehensive comparison between models and offer an integrated perspective on performance in relation to both metrics, we introduce the F*R/100 measure. This measure is calculated as the product of FID and ICR, serving as a unified metric for evaluating model performance.

## 5.2 COMPARISON WITH PREVIOUS METHODS

As illustrated in Table 2, GEOM-ERASING notably surpasses existing erasure methods across three distinct implicit concepts by substantially diminishing their occurrence in the synthesized images. Even in instances where initial images all contained text, our method remarkably reduces text presence to just 13.48% of the images, thus significantly minimizing the generation of unintended concepts. In addition to reducing unintended elements, GEOM-ERASING also improves the overall quality of the images compared to other methods like SD (Rombach et al., 2022) and different baseline models. This improvement in image quality is due to the method's ability to effectively erase implicit concepts—since the ideal images (ground truth) don't contain these elements, avoiding them results in higher quality scores (FID scores). Further insights into the correlation between erasure efficacy and enhanced image quality are detailed in our ablation study in Sec. 5.3.

The methods of FMN, NP, and SLD demonstrate limitations in effectively removing implicit concepts. The performance of these methods relies on the diffusion model's capability to identify specific concepts as discussed in Sec. 2. However, identifying implicit concepts is a notable challenge for these models. This challenge is underscored by the attention map images provided in Appendix B, which depict the models' inadequacies in accurately identifying and addressing implicit concepts, subsequently hindering successful erasure.

Among all the baselines, our observations indicate that ESD (Gandikota et al., 2023) demonstrates superior erasure performance, albeit with a higher FID score. This can be attributed to the approach employed by ESD, where the fine-tuned SD model is trained to move away from the images it originally generated, regardless of whether they contain the intended concept or not. However, since original generated images may contain implicit concepts with high probability, ESD might result in unintended concept removal while affecting meaningful one, as in the 2nd column of Fig. 7.

Table 3: **Ablation study of GEOM-ERASING components.** Merely appending concept names to the original text conditions proves insufficient. The geometric component plays a crucial role, while the reweighted loss optimizes generation quality, exhibiting negligible impact on the ICR.

| Concept | Geometric | Loss reweight | FID | ICR | F*R |
|---------|-----------|---------------|-----|-----|-----|
|  |  |  | 7.59 | 30.40 | 230.74 |
| √ |  |  | 7.06 | 17.04 | 120.30 |
|  | √ |  | 6.97 | 11.18 | 77.92 |
|  |  | √ | 6.46 | 29.38 | 189.79 |
| √ | √ |  | 6.81 | 7.36 | 50.12 |
| √ | √ | √ | **6.42** | 7.23 | **46.42** |
| *Fine-tuning with 0% watermark (oracle)* |  |  | 6.93 | **7.13** | 49.41 |

In contrast, our proposed method, GEOM-ERASING, demonstrates the ability to effectively remove implicit concepts while preserving the other meaningful concepts, yielding favorable ICR and FID results. It surpasses the state-of-the-art, as evidenced by the superior F*R/100 measure. Refer to Fig. 1 for visual comparisons between different methods. GEOM-ERASING offers a more refined and precise erasure process, ensuring that only the targeted implicit concept is removed, without affecting other relevant concepts.

## 5.3 ABLATION AND ANALYSIS

In this section, we perform various ablations and hyper-parameter analyses to demonstrate the effectiveness of different components in GEOM-ERASING. Additionally, we investigate the impact of geometric accuracy on the overall performance of our method, considering its inherent dependence on this aspect. Furthermore, we explore integrating our method with Negative Prompt to showcase its compatibility and potential synergies.

**Ablative analysis.** The ablation results, as shown in Table 3, shed light on the importance of different components in our methods. We sequentially integrated the concept name (Eq. 2), geometric information (Eq. 3), and loss reweight (Eq. 4) before combining them. Notably, the geometric component proved pivotal, markedly reducing both FID and ICR, particularly when synergized with the concept condition, enhancing the model's overall performance. The loss reweight component contributes to improving the visual appeal of the generated images while maintaining the efficacy of implicit concept removal. Throughout the ablative studies, a consistent trend between FID and ICR is observed, implying enhanced erasure correlates to superior image quality.

Moreover, when fine-tuning the model with no implicit concepts (0% images with watermarks), the model, in its optimal state, achieves an ICR comparable to GEOM-ERASING. Interestingly, GEOM-ERASING surpasses even this optimum in FID and F*R, emphasizing the importance of geometric information in refining concept learning and subsequently improving image quality.

**Choice of bin size, number of selected bins and reweight loss.** The choice of bin size, the number of bins, and reweight loss will influence erasure outcomes. Table 4 depicts the effects of varying bin sizes $M$ and selected bin numbers $K$, with bold values denoting optimal performance and the gray row signifying the default selection. Bins are ranked and selected by value $p_i$ as stated in Sec. 4. Initially, we analyze the impact of bin sizes by fixing the ratio between the number of selected bins and bin size and varying the size from $8^2$ to $64^2$. As the bin size increases, the performance initially improves and then starts to decline. This trend suggests that higher resolutions may provide more accurate concept localization but can also dilute the information density of the original text. Subsequently, with a fixed bin size, varying the number of selected bins showed enhancement in erasure performance up to a saturation point.

For the reweight loss, we conducted ablation experiments based on the model in the gray row of Table 4. An alternate reweight loss incorporating the $p_i$ values was proposed. As shown in Table 5, applying the reweight loss leads to improved FID compared to the model without the loss. However,

Table 4: **Comparisons between different choice of Bin size and select bins.**

| M | K | FID | ICR | F*R |
|---|---|-----|-----|-----|
| $8^2$ | 4 | 6.78 | 18.26 | 123.80 |
| $16^2$ | 16 | 6.53 | 15.45 | 100.89 |
| $32^2$ | 64 | **6.51** | 12.64 | 82.29 |
| $64^2$ | 150 | 7.29 | 16.47 | 120.07 |
| $32^2$ | 72 | 6.81 | **7.36** | **50.12** |
| $32^2$ | 80 | 6.92 | 7.35 | 50.86 |

Table 5: **Comparison between different designs of the reweight loss.**

| Re-weight Function | $\alpha$ | FID | ICR | F*R |
|---|---|-----|-----|-----|
| Eq. 5 | 0.25 | 6.42 | **7.23** | **46.42** |
| | 0.50 | 6.40 | 7.63 | 48.83 |
| | 0.75 | 6.45 | 7.41 | 47.79 |
| $(1-p_i)^\alpha$ | 0.5 | 6.27 | 9.21 | 57.75 |
| | 1 | 6.33 | 9.34 | 46.46 |
| | 2 | **6.26** | 9.44 | 59.09 |

| Negative Prompt | FID | ICR | F*R |
|---|-----|-----|-----|
| w/o NP | 6.42 | 7.23 | 46.42 |
| Concept | 6.15 | 7.03 | 43.23 |
| Uniform Geometry | 6.99 | 5.02 | 35.09 |
| Random Geometry | 7.12 | 4.98 | 35.46 |

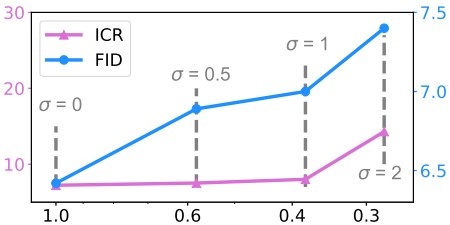

Figure 4: **Results under different geometric accuracy with varying noise magnitudes** $\sigma$. The X-axis represents the geometric accuracy.

Table 6: **Adopting the implicit concept in the negative prompt.** Usage of geometry improves ICR at the cost of worse FID.

utilizing the $p_i$ may degrade the erasure performance. Opting for simplicity and effectiveness, a fixed value was utilized for the area covered by the implicit concept, as outlined in Eq. 5.

**Geometric accuracy.** We execute experiments to investigate the sensitivity of GEOM-ERASING to the precision of geometric information provided. Upon selecting the bins, we introduce two noise scalars, $\epsilon_1, \epsilon_2 \sim \mathcal{N}(0, \sigma^2)$, to the selected index, illustrated as $y' = y \oplus y_{\text{im}} \oplus \langle l\{m + \epsilon_1, n + \epsilon_2\}\rangle$. Variations in $\sigma$ yield distinct IoU values between the originally selected and noised bins, visualized in Fig. 4. The ICR can tolerate a geometry accuracy up to 0.4 IoU; however, the erasure performance experiences a decline as accuracy continues to decrease.

**Negative prompt.** We aim to improve our method, GEOM-ERASING, by adding both the learned concept name and geometric information to the negative prompt to better erase unwanted details. We test using location tokens that are picked both uniformly and randomly to represent geometric information. As shown in Table 6, adding the concept name to the negative prompt improves how well unwanted details are erased and the overall quality of generated content due to the model's improved ability to recognize concepts. Adding geometric information, whether uniformly or randomly, further improves the erasure, but it also tends to increase the FID. We plan to explore the reasons for this increase in more detail in future work.

## 6 CONCLUSION

Fine-tuning on personalized datasets is a prevalent practice, but the presence of unwanted implicit concepts like QR codes, watermarks, and text within these datasets can pose significant challenges during the refinement of personal diffusion models. This paper delves into the substantial impact of such implicit concepts, establishing a formal framework for their removal. Conventional methods, which predominantly depend on pre-trained diffusion models or merely acknowledge concept existence, falter in eradicating these implicit elements. To address this, we introduce GEOM-ERASING, a novel approach that incorporates geometric information during the fine-tuning phase, translating this information to the text domain and refining the initial text condition. We substantiate our approach through three diverse datasets, each laden with distinct implicit concepts. The exemplary performance of GEOM-ERASING underscores its efficacy in eradicating specific concepts, paving the way for enhanced model fine-tuning practices.

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

IC-QR IC-Watermark IC-Text

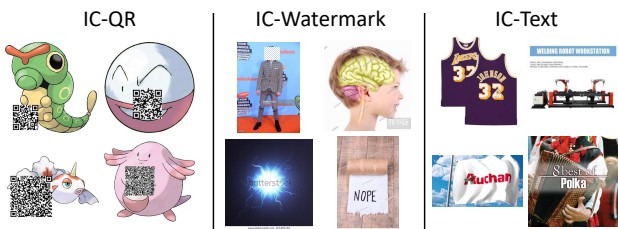

Figure 5: **Image samples from our Implicit Concept (IC) dataset.**

APPENDIX

## A    DATASETS DETAILS

**IC-QR.**    Real QR codes to the Pokemon dataset (Pinkney, 2022). The total dataset contains 802 image-text pairs, which is divided into two portions: 80% for fine-tuning, and the remaining 20% for testing. In the training subset, QR codes are pasted to 25% of the images, with QR code lengths varying from 1/4 to 1/2 of the image length, placed randomly, occasionally overlapping with the original content to resemble real-world scenarios. Importantly, test images remain QR code-free for evaluation. To provide concept conditions and geometric information for our method and evaluation, a Faster-RCNN detector is trained using an open-source QR detection dataset[3].

**IC-Watermark.**    Images are collected from CC12M (Changpinyo et al., 2021), amounting to 320k images, with half containing watermarks. A watermark recognition tool trained by LAION is employed to identify watermarked images from CC12M with a high confidence threshold of 0.9 to ensure accuracy. For preliminary experiments, subsets of 160k images with varying ratios of watermarked images are constructed. In other experiments, a consistent dataset of 80k images with watermarks and 80k images without watermarks is selected. To provide concept conditions, the watermark recognition tool is used, and for geometric information, the classifier activation map produced by the tool is employed, deciding areas of containing watermarks.

**IC-Text.**    Text images are gathered from LAION (Schuhmann et al., 2021). The training dataset we used is provided by (Yang et al., 2023), known as LAION-Glyph. It comprises 1M samples, with each image containing text. For the evaluation dataset, 2k text-free images are collected. To obtain geometric information and for evaluation purposes, PP-OCRv3 Du et al. (2021) is used to detect text within the images.

## B    LIMITATION OF BASELINE MODELS.

As discussed in Sec.5.2, methods like FMN, NP, and SLD heavily rely on the model's capacity to recognize specific concepts. Following our experiments in ablation, we provide the "watermark" concept as an illustration here to check the cross attention scores to see this recognition ability. It's important to note that NP and SLD operate during the sampling phase, thus depending on the original SD's capabilities. As shown in Fig. 6, SD, NP, SLD, and FMN do not exhibit attention to the regions containing the ground truth watermark. In contrast, GEOM-ERASING successfully directs attention to the meaningful areas that encompass the watermark. This indicates that learning implicit concepts clearly can help erase them.

ESD exhibits superior erasure results compared to other baseline methods; however, it does come at the expense of image quality. This phenomenon is attributed to ESD's tendency to steer generation away from the fine-tuned SD model. While this strategy reduces the occurrence of unwanted implicit concepts, it also affects the generation of other valuable content. As a result, ESD generates fewer images containing implicit concepts but produces images that differ from those generated by other baseline methods, as illustrated in row two, columns 2 and 6 of Fig. 7.

---

[3]https://universe.roboflow.com/roboflow-qsmu6/qr-codes-detection

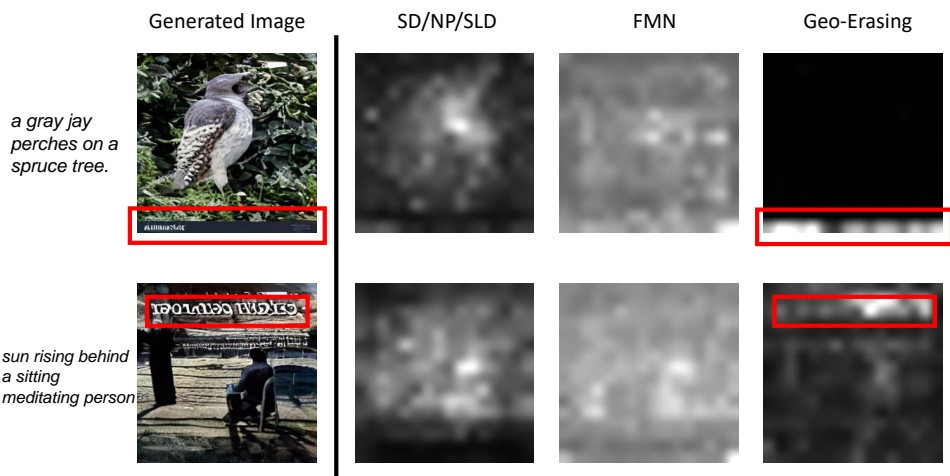

Figure 6: **Visualization of the cross attention map.** The left column is the generated image. Each column in the right part is the cross-attention map between the concept 'watermark' and the image of each method. It can be seen that GEOM-ERASING can attend to the location with watermark, while other methods can not. Since the recognition ability of SD/NP/SLD is poor, they cannot erase the implicit concept.

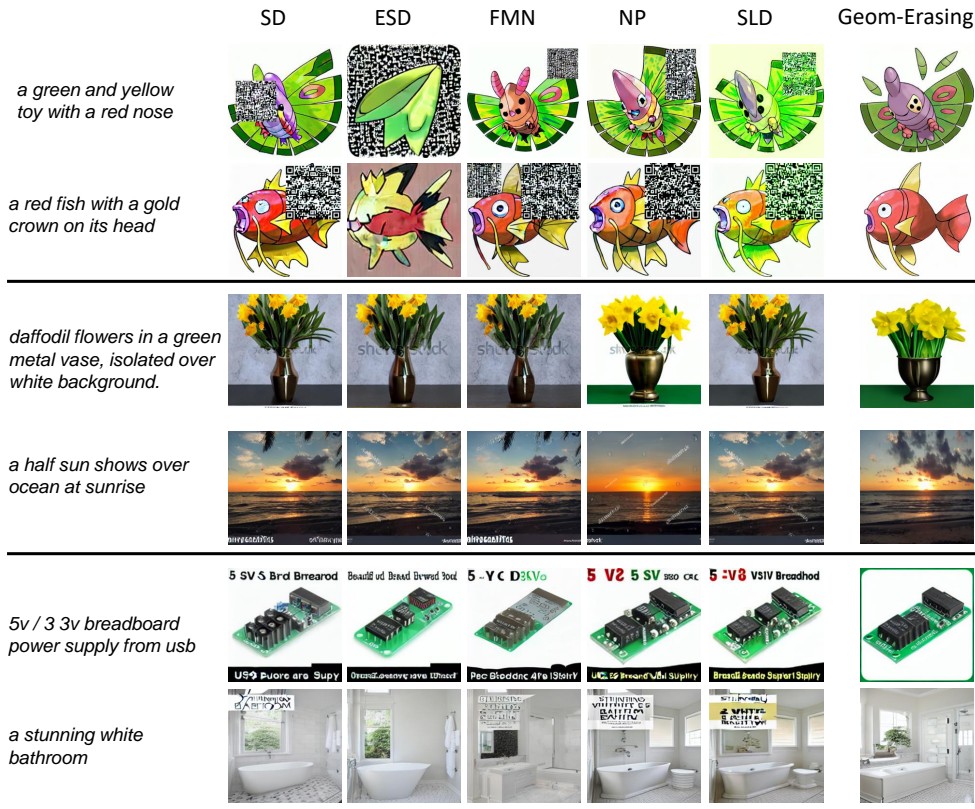

Figure 7: **More generation examples.** The first group of images are fine-tuned on IC-QR. The middle and the bottom are fine-tuned on IC-watermark and IC-Text, respectively.

