# OpenReview forum: "Geom-Erasing: Geometry-Driven Removal of Implicit Concept in Diffusion Models"
_ICLR.cc/2024/Conference — ICLR 2024 Conference Withdrawn Submission_

### Official Review · Reviewer_3V6v · 2023-10-22

**Soundness:** 2 fair
**Presentation:** 3 good
**Contribution:** 2 fair
**Rating:** 5
**Confidence:** 3

**Summary:**

The paper introduces an approach to erase concepts in images generated by diffusion models. The paper considers undesired concepts common in images like watermarks, QR codes, and text. The application of this method is useful in scenarios when a user wants to fine-tune a diffusion model on a private dataset that has a lot of undesired concepts like watermarks. To tackle this, problem the paper introduces Geom-Erasing, which has the following components: a concept detector/localizer, a text editing approach, and a mask-based fine-tuning objective. The experimental results show that the proposed method has significantly better performance than existing approaches.

**Strengths:**

1. The paper is well-written with good visualizations showcasing the efficacy of their approach.
2. Geom-erasing achieves significantly better results than baseline approaches. The heuristical approach is quite useful when a user is utilizing current pre-trained models on small-scale datasets with known concepts.
3. The paper provides extensive ablations to evaluate the efficacy of their approach.

**Weaknesses:**

1. The technical novelty of the proposed approach is limited. The improvement in concept erasure comes from text prompt editing and fine-tuning with the mask-based objective. The justification behind using the mask-based objective is not well supported. Specifically, it relies on the fact that the pre-trained model (SD) is not biased toward generated undesired concepts like watermarks. Therefore, when you relax the loss function on certain parts of the image it doesn't overfit on the fine-tuned dataset and lets the pre-trained model generate without constraints.  But what if the pre-trained model is biased (e.g., generates more watermarks) in itself? Will this model work? This can be tested by training on a separate watermark dataset, before fine-tuning on the desired downstream dataset.
2. The claim of erasing "implicit concepts" is arguable. The current method heavily relies on an object detector that detects the exact position of an object like watermarks, texts, etc. I would assume concepts that would be implicit in an image are lighting conditions (sunny or cloudy), facial expressions (whether a person is angry or happy), etc. As it is hard to locate such features in an image it would be hard to extend the current approach for such applications. Given this, I feel the claims in this paper should be toned down to object erasure.
3. The paper needs to present results using multiple diffusion models to thoroughly evaluate its efficacy.
4. It would be interesting to see how the original ICR in the fine-tuned dataset affects the performance of Geom-erasing.

Minor comments:

Section 3.2: change "We" -> "we"

Section 3.2: earlier on in the paper it is unclear how the watermarks are detected. It would be good have some of these details before presenting the method.

Line before Eq. 3, what is l{., .}?

**Questions:**

Please refer to the weakness section.

---

### Official Review · Reviewer_GREc · 2023-10-31

**Soundness:** 3 good
**Presentation:** 4 excellent
**Contribution:** 2 fair
**Rating:** 5
**Confidence:** 3

**Summary:**

The authors bring in a new watermark-erasing approach called GEOM-ERASING. They use a watermark classifier from LAION to extract geometric information during the fine-tuning phase and encode this information to the text domain by adding location tokens and refining the initial text inputs. They also introduce a new concept called 'implicit concept removal' and construct three datasets termed 'Implicit Concept'.

**Strengths:**

1. The paper created three distinct datasets called 'Implicit Concept', each containing specific implicit concepts such as watermarks, QR codes, and text. This diversity enables comprehensive testing and evaluation of the paper's method in various contexts and is highly valuable for future research. Additionally, this diversity contributes to validating the method's universality, ensuring its effectiveness in various applications.

2. The model is able to omit the unintended implicit concepts after fine-tuning via their methods. The method proposed in the paper has been demonstrated to be effective on the datasets, successfully freeing image generation from implicit concepts.

3. The paper is meticulously crafted with clear and concise language, ensuring that the reader can easily grasp the information presented. Also, the content within the paper is thorough and all-encompassing, leaving no important aspect unaddressed. The experiments are abundant and prove the effectiveness of their methods.

**Weaknesses:**

1. The concept of 'implicit concepts' is vague. Watermarks, QR codes, and text are the same and can be categorized as watermarks in my opinion.
2. What you've done seems more like a bridge between a stable diffusion model and an image classifier model, lacking in innovation. Also, in consideration of this, GEOM-ERASING should not be compared to those concept-erasing methods built inherently in the diffusion models.
3. It is nice to see you mention the ICR and FID trade-off in the preliminary study, which is the biggest concern when I read the abstract. However, I haven't seen a solution to this problem, which also diminishes the value of the proposed method.

**Questions:**

1. In section 2: Concept erasing in diffusion models, you mention 'Current methods to erase unwanted concepts mainly depend on the model’s ability to recognize those concepts'.
2. How to achieve a balance between ICR and FID? How do other methods perform in addressing this issue?

---

### Official Review · Reviewer_dnxP · 2023-11-01

**Soundness:** 2 fair
**Presentation:** 2 fair
**Contribution:** 2 fair
**Rating:** 5
**Confidence:** 4

**Summary:**

This paper introduces GEOM-ERASING, an innovative approach that fine-tunes diffusion models using personalized datasets and effectively eliminates unintended concepts like watermarks and QR codes. The method integrates an auxiliary module, such as a classifier or detector, to capture the geometric attributes of these concepts and translates them into the text domain. This alignment ensures that these attributes correspond with the text conditions of the diffusion models. During testing, the model generates samples devoid of these undesired concepts by excluding the associated text inputs. Experimental outcomes highlight GEOM-ERASING's capability not just in identifying but also in precisely removing specific implicit concepts.

**Strengths:**

- The proposed method is overall simple but effective. It does not require additional information from the dataset, except for an additional module in detecting the undesired concepts, which can be plugged-in using existing pre-trained models. The application discussed in the paper can be useful to practioners.

- The paper is overall organized, facilitating a smooth reading experience. Additionally, the inclusion of model overviews and illustrative figures for the components simplifies the understanding of the proposed method.

- The experimental results well validate the approach. Notably, the paper also provides an ablation study to interpret the effects of the hyperparameters.

**Weaknesses:**

- In 3.1, the description of the latent space where the diffusion model is operated, appears ambiguous. The authors mention training the diffusion models within a VQ (quantized) space, traditionally recognized as a discrete domain. However, there's an indication that a Gaussian diffusion model is employed over this discrete space following the remaining statement of the paper. The methodology behind rounding continuous predictions to yield discrete outcomes remains unclear. If the latent space differs from being a VQ space, it's necessary that the description be revised, accompanied by detailing the construction of the latent space.

- As the method embed the geometric information in a textual form, it is non-trival to explore deeper how well the text presents these geometric information. The visualization visualizes the attention map between the watermark and the image, but it does not show the correlation to the text embedding. Some additional may be benefitial to justify this point. For example, if we add "watermark <another location>" into the text, can the model move the watermark from the original place to another place? If so, it will provide better interpretation of the method and make the method more convincing.

- The method is closely related to Dreambooth (Ruiz et al., 2023). The authors may need to carefully discuss the relation including connections and differences between this paper and the works related to Dreambooth. However, the discussion related to this point is not elaborated in the paper.

- While the primary scope of the paper centers on the removal of implicit concepts with geometric attributes, it would be enriching to explore the method's efficacy in eliminating explicit concepts, such as objects. Alternatively, a discussion on the similarities and distinctions from prior works in this direction would provide valuable context and depth to the paper.

- In the ablation study, the authors may also include the impact of the concept detecting to provide a study regarding the robustness of the proposed method.

----------

*Given the aforementioned concerns, my current inclination is to rate this submission below the acceptance threshold. However, I am open to reconsidering my evaluation if the authors address the highlighted issues or provide clarifications in their rebuttal.*

**Questions:**

Please see the Weakness.

---

### Official Review · Reviewer_ci9W · 2023-11-02

**Soundness:** 3 good
**Presentation:** 3 good
**Contribution:** 1 poor
**Rating:** 3
**Confidence:** 2

**Summary:**

This paper proposes GEOM-ERASING algorithm, Using removes the multimodal diffusion model to eliminating specific implicit concepts of watermarks, QR codes, and text with geometric method. The experimental results show the effectiveness of identifying and removing specific implicit concepts taking advantage of the contextual information.

**Strengths:**

1.Using geometric method to erase the specific implicit concepts, i.e., watermarks, QR codes, and text.
2.Using the context information to help encode geometric information of implicit concepts.
3.The algorithm obtains good performance.

**Weaknesses:**

1.The contribution is limited. It seems that the method is built upon several existing techniques, and the insight of removing geometric concepts is trivial.
2.The method is limited in the specific type of implicit concepts, i.e. the geometric concepts. It seems that such geometry-oriented method is hard to generalized to removing pixel level concepts, such as adversarial perturbations or corruptions.

**Questions:**

1.What is key challenge of the using the geometric method removing the implicit concepts, is that trivial and straightforward?
2.Except for geometric concepts, can such method still effective on pixel level perturbation?